# VQ-learning: Towards Unbiased Action Value Estimation in Reinforcement Learning

## Abstract

Q-learning as a well-known Reinforcement Learning algorithm is prone to overestimation of action values in stochastic settings. Such an overestimation is mainly due to the use of the max operator when updating the Q function. Deep Q-learning (DQN) suffers from the same problem which is further aggravated by noisy learning environment, and can lead to substantial degradation of reward performance. In this work, we introduce a simple yet effective method called VQ-learning, along with the extended version using function approximation, called Deep VQ-Networks (DVQN), which regulates the estimation of action values and effectively tackles the issue of biased value estimation. While Double Q-learning has been proposed to tackle the same issue, we showcase that VQ-learning provides better sample efficiency, even when the overestimation bias preconditions are eliminated. We also evaluate DVQN on Atari-100k benchmark and demonstrate that DVQN consistently outperforms Deep Q-learning, Deep Double Q-learning, Clipped Deep Double Q-learning, Averaged DQN and Dueling Deep Q-learning in terms of reward performance and sample efficiency. Moreover, our experimental results show that DVQN serves as a backbone network better than DQN, when combined with an additional representation learning objective.

## 1 Introduction

Q-learning (Watkins, 1989) has been one of the most widely applied reinforcement learning algorithms. Combined with function approximation through neural networks, deep Q-learning (DQN) (Mnih et al., 2015) shows strong adaptability to complex discrete control tasks. However, Q-learning is known to suffer from overestimation bias (Thrun & Schwartz, 1993; Hasselt, 2010) due to the maximization operation when computing the temporal difference target. When the overestimation of the Q-function is uniform over all actions, the relative preferences among action values remain unchanged, and therefore the policy remains the same. However, when overestimations are not uniform, they can be detrimental to policy learning Thrun & Schwartz (1993). This phenomenon is particularly problematic with deep Q-learning (DQN) Van Hasselt et al. (2016).

One classic solution to the problem of overestimation is Double Q-learning (Hasselt, 2010). It introduces two independent unbiased Q-functions to estimate action values. To update the first Q-function, action with the highest value associated with Q-function one is selected and evaluated by the Q-function two, it is just the opposite when updating the second Q-function. Double Q-learning is guaranteed to underestimate the maximum expected action values. Deep Double Q-learning (DDQN) (Van Hasselt et al., 2016) extends this idea to the setting of deep Q-learning, in that it treats the target Q-function in DQN as the second independent unbiased action value estimator. DDQN has been shown to alleviate the overestimation problem and improve the reward performance in a wide range of domains. However, the target Q-function in DQN is introduced to stabilize the training process and is synchronized with the Q-function regularly. Therefore, the two Q functions in DDQN are not fully independent, in some cases DDQN still suffers from overestimation. Clipped Double Q-learning (CDDQN) (Fujimoto et al., 2018) addresses this concern by taking the minimum between the two independent Q-functions in an actor-critic setting (Silver et al., 2014). CDDQN is applied to tackle overestimation in TD3 (Fujimoto et al., 2018) and SAC (Haarnoja et al., 2018), two of the most popular RL algorithms mainly for continuous control tasks. However, CDDQN is still not a complete answer,

it can effectively suppress the overestimation bias, but also tends to underestimate at the same time, since the employed minimum operation over action value evaluation is not lower bounded. Moreover, Anschel et al. (2017) propose another solution to the overestimation phenomena, called Averaged DQN, which averages across previously learned Q values estimates to compute the target Q value. Averaged DQN helps reduce the variance of target approximation error, resulting in improved performance and stability.

In this work, we introduce a novel RL approach named VQ-learning, to tackle the biased value estimation while enhancing sample efficiency. Compared to Q-learning, VQ-learning incorporates an additional independent and unbiased state value function, which is used to compute the target values for the Q function. We also propose Deep VQ-learning, a deep extension of VQ-learning (DVQN), to handle complex domains. DVQN employs a separate network to approximate the additional state value function, which can be used to regulate over- and underestimation biases of Q values through a knowledge distillation loss (Hinton et al., 2015). With experimental results, we demonstrate that DVQN outperforms DQN, DDQN, CDDQN and Averaged DQN in terms of reward performance. Additionally, we offer a convergence proof for VQ-learning, supported by an empirically validated condition.

By employing an independently learned state value function to regularize the estimation of the Q-function, DVQN not only addresses biased value estimation effectively, but also directly enhances sample efficiency of learning at the same time. From this perspective, DVQN shares a few similarities with Dueling DQN (Wang et al., 2016), another well-known approach aimed at improving sample efficiency, which also involves learning a state value function, albeit not independently (for more analysis, see section 4.1). Despite the efficacy of Dueling DQN, we show that our proposed approach DVQN outperforms Dueling DQN in Atari game benchmark.

Furthermore, another crucial strategy towards improving sample efficiency of RL algorithms employed by several works is to improve state representations by employing explicit representation learning objectives. For example, Yarats et al. (2021) propose using the reconstruction loss as an auxiliary loss alongside the RL loss. Zhang et al. (2020) learn an invariant representation without reconstruction. In Laskin et al. (2020) and Stooke et al. (2021), momentum contrastive learning objectives are imposed on RL algorithms to improve the performance. We demonstrate that when using additional representation learning objectives, DVQN serves as a better backbone architecture than DQN. In particular, we apply a *temporal contrastive learning objective* (which enforces similar representations for temporally closer states) to the feature extraction module of DVQN. In our experimental results, DVQN combined with the temporal contrastive objective shows superior reward performance in comparison with combining the same objective with DQN. More details regarding the temporal contrastive objective we apply can be found in Appendix A.6.

To summarize we make the following main contributions.

1. We propose a novel paradigm of VQ-learning and further develop Deep-VQ Networks (DVQN) to tackle the biased estimation of action values and improve sample efficiency of learning. DVQN outperforms our baselines in 5 arbitrarily chosen Atari domains.

2. Our experiments reveal that, when using additional representation learning objectives like temporal constrastive loss, DVQN serves as a better backbone architecture than DQN.

3. Our method (DVQN) is easy to implement and requires limited additional computational cost compared to DQN. We release the corresponding source code at `https://anonymous.4open.science/r/DVQN-046F`.

## 2 Background and Related Work

Reinforcement learning (RL) involves the interaction between the agent and the environment. An RL task can be depicted by a Markov Decision Process (MDP) in form of a tuple $\mathcal{M} = \langle \mathcal{S}, \mathcal{A}, \mathcal{P}, \mathcal{R} \rangle$, where $\mathcal{S}$ and $\mathcal{A}$ indicate the state and action space respectively, $\mathcal{P}(s'|s, a)$ and $\mathcal{R}(s, a)$ denote transition probability function and the reward function. The goal is to learn a policy $\pi(s)$ that maximizes the expected discounted cumulative reward for any given state. Given a policy $\pi$, we can define the discounted accumulative reward when taking

action $a$ on state $s$:

$$Q_\pi(s,a) = \mathbb{E}_\pi \left[ \sum_{k=0}^{\infty} \gamma^k R_{t+k+1} \mid S_t = s, A_t = a, \pi \right] \tag{1}$$

where $\gamma$ is the discount factor. The goal is to find the optimal value function $Q^*(s,a) = \max_\pi Q_\pi(s,a)$ so that the optimal policy can be induced by :

$$\pi^*(a|s) = \begin{cases} 1 & \text{if } a = \underset{a \in \mathcal{A}}{\mathrm{argmax}} Q^*(s,a) \\ 0 & \text{otherwise} \end{cases} \tag{2}$$

$Q^*(s,a)$ is also the unique solution of Bellman optimality equation as: $Q^*(s,a) = R(s,a,s') + \gamma \sum_{s' \in \mathcal{S}} \mathcal{P}_{ss'}^a \max_{a'} Q_*(s',a')$. Q-learning is a classic model-free RL algorithm to compute $Q^*(s,a)$, which updates $Q(s,a)$ iteratively by:

$$Q(s,a) = Q(s,a) + \alpha \left[ R(s,a,s') + \gamma \max_{a'} Q(s',a') - Q(s,a) \right] \tag{3}$$

In Deep Q-learning (DQN) (Mnih et al., 2015), $Q(s,a)$ is approximated by a neural network $\phi$, which is iteratively updated by minimizing the loss function below using gradient descent:

$$\mathcal{L}(\phi) = \left[ r + \gamma \max_{a' \in \mathcal{A}} Q_{\hat\phi}(s',a') - Q_\phi(s,a) \right]^2 \tag{4}$$

where $(s,a,s',r)$ is sampled from the replay buffer which stores a fix amount of previous experiences dynamically. $\hat\phi$ denotes the parameters of the target Q network which is synchronized with $\phi$ at regular intervals. To stabilize the training of $Q_\phi$, regular synchronization of its parameters with that of the target network is crucial. Considering the stochasticity and noise induced the neural network function approximator, DQN is prone to overestimation of Q values (Van Hasselt et al., 2016) when evaluating the selected action $a'$, due to the max operator.

To tackle the overestimation, Double DQN (DDQN) (Van Hasselt et al., 2016) takes the target network $Q_{\hat\phi}(s,a)$ in DQN as another unbiased action value estimator to evaluate action $a'$ selected by $Q_\phi(s,a)$, thus its loss function becomes:

$$\mathcal{L}(\phi) = \left[ r + \gamma Q_{\hat\phi}(s', \arg\max Q_\phi(s',a')) - Q_\phi(s,a) \right]^2 \tag{5}$$

However, since $\hat\phi$ is synchronized with $\phi$ regularly, $Q_\phi(s,a)$ and $Q_{\hat\phi}(s,a)$ are not fully independent. Thus, the overestimation cannot be effectively alleviated.

Clipped Double DQN (CDDQN) (Fujimoto et al., 2018) improves over DDQN by employing two independent action value estimators: $Q_{\phi_1}(s,a)$ and $Q_{\phi_2}(s,a)$. CDDQN was first introduced in an actor-critic setting, it can be adapted in a critic-only setting by ensuring that the policy which interact with the environment is induced by $Q_{\phi_1}(s,a)$. When computing the target value, action $a'$ with the highest value on $s'$ is selected according to $Q_{\hat\phi_1}(s',a')$ and then evaluated by the minimum between $Q_{\hat\phi_1}(s',a')$ and $Q_{\hat\phi_2}(s',a')$. The loss function optimized by CDDQN is given as:

$$\mathcal{L}(\phi_i) = \left[ r + \gamma \min_{i \in \{1,2\}} Q_{\phi_i} \left( s', \arg\max_{a'} Q_{\hat\phi_1}(s',a') \right) - Q_{\phi_i}(s,a) \right]^2 \tag{6}$$

While CDDQN overcomes the overestimation problem it is still prone to underestimation (Ciosek et al., 2019), since the target value is not lower bounded. Improving over both DDQN and CDDQN, our approach tackles simultaneously over- and under-estimations by introducing a novel regularization term based on a independent value function approximation for states, which is used in a knowledge distillation framework (Hinton et al., 2015) to regulate the Q-values.

Average DQN (Anschel et al., 2017) also aims at solving the overestimation issue by averaging previously learned Q values estimates, more specifically, averaging across previous versions of target Q networks. The averaged target Q value is used to replace the original target Q value in the DQN loss function.

Instead of aiming at tackling overestimation, Dueling DQN (Wang et al., 2016) directly focuses on improving sample efficiency of DQN to improve reward performance. With Dueling DQN, the single-stream Q network is converted into a combination of two streams of networks, that are supposed to approximate the state value function and advantage value function respectively. Dueling DQN utilize the fact that in complex domains, there are many states where action choices are task-irrelevant. With the Dueling architecture, each update of Q values will update the stream for state values via back propagation. This in turn updates the value estimations for all actions available on the same state, thus improving sample efficiency of learning. In contrast, with a single-stream architecture such as DQN, only the value corresponding to one action can be updated at a time.

## 3 VQ-learning

We propose VQ-learning, a novel reinforcement learning algorithm which addresses the overestimation bias of Q-learning and improves sample efficiency at the same time.

Unlike traditional Q-learning, VQ-learning involves learning separate state and action value functions, denoted by $V$ and $Q$, respectively, which jointly contribute to policy evaluation, as detailed in Algorithm 1. VQ-learning inherently avoids overestimation bias since its state and action value functions are updated without employing the maximum operation.

VQ-learning can be understood as a variant of generalized policy iteration (GPI) (Sutton et al., 1998), which involves interaction between policy iteration and policy improvement. Policy evaluation can be decomposed into two stages of TD updates: the state value function $V$ is updated for the current policy, followed by the update of the action value function $Q$ using $V$ to compute TD-target (see line 9 and 10 in Algorithm 1). Subsequently, policy improvement according to the updated $Q$ function generates a new $\epsilon$-greedy policy (line 6 in Algorithm 1).

In section 3.1, we compare VQ-learning against Double Q-learning and Q-learning across three distinct task configurations. In section 4, we introduce Deep VQ Network, an extension of VQ learning to solve tasks in more complex domains.

---

**Algorithm 1** VQ-learning

---

1: Initialize $Q(s, a)$ arbitrarily for all $s \in \mathcal{S}, a \in \mathcal{A}$, except $Q(teminal, \cdot) = 0$
2: Initialize $V(s)$ arbitrarily for all $s \in \mathcal{S}$, except $V(terminal) = 0$
3: Initialize step size $\alpha$ and $\epsilon \in (0, 1)$
4: **for** for each episode **do**
5:     Initialize S
6:     choose $a$ from $s$ using policy derived from $Q(s, a)$ ($\epsilon$-greedy)
7:     **repeat** for each step of episode
8:         Take action $a$, observe $s'$, $r$
9:         $V(s) \leftarrow V(s) + \alpha[r + \gamma V(s') - V(s)]$
10:        $Q(s, a) \leftarrow Q(s, a) + \alpha[r + \gamma V(s') - Q(s, a)]$
11:        $s \leftarrow s'$,
12:     **until** $s$ is terminal

---

By the following theorem, we guarantee the convergence of tabular VQ learning under four conditions.

**Theorem 1** (Convergence of VQ-learning). *Consider a finite state-action MDP and apply GLIE (i.e. non-greedy actions are chosen with vanishing probabilities) learning policy $\pi$ given as a set of probabilities $\Pr(a|s, t, n_t(s), Q)$. Assume that at time step $t$, action $a_t$ is chosen according to $\pi$ which uses $Q = Q_t$. $V_t$ and $Q_t$ are updated to $V_{t+1}$ and $Q_{t+1}$ as follows:*

$$V_{t+1}(s_t) = (1 - \alpha_t) V_t(s_t) + \alpha_t (r_t + \gamma V_t(s_{t+1})) \tag{7}$$

$$Q_{t+1}(s_t, a_t) = (1 - \alpha_t) Q_t(s_t, a_t) + \alpha_t (r_t + \gamma V_t(s_{t+1})) \tag{8}$$

*$Q_t$ converges w.p.1 to the optimal Q-function as long as:*

1. *Q and V values are stored in lookup tables.*

2. $\mathrm{Var}[r(s,a)] \leq \infty$

3. $0 \leq \alpha_t \leq 1, \quad \sum_t \alpha_t(x,a) = \infty, \quad \sum_t \alpha_t^2(x,a) < \infty$

4. $\lim_{t\to\infty} V_t(s) = \lim_{t\to\infty} Q_t(s, a_t)$, $a_t = \pi^{Q_t}(s)$, *where* $\pi^{Q_t}(s)$ *is the GLIE policy derived from* $Q_t$

We refer the proof to Appendix A.1.

### 3.1 VQ-learning in different task settings

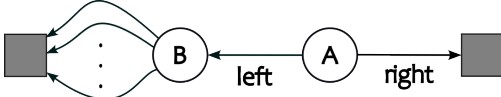

**Figure 1:** Example task to illustrate the effectiveness of VQ-learning. The gray colored boxes represent terminal states.

**Stochastic rewards.** To illustrate the effectiveness of VQ-learning in mitigating overestimation, we compare it with Q-learning (Watkins & Dayan, 1992) and Double-Q learning (Hasselt, 2010) using a simple yet intuitive RL task with stochastic rewards. This task is also known to identify the effectiveness of Double Q-learning against overestimation (Sutton et al., 1998).

As shown in Figure 1, each episode starts from state A, which has two available actions. Taking right transitions to a terminal state with a reward of zero. Taking left transitions to state B, where 20 actions are available, each action leads to the other terminal state and leads to a reward sampled from Gaussian distribution $\mathcal{N}(-0.1, 1)$. Thus, the expected return of taking left action is $-0.1$, implying that the optimal policy is to take right action on state A and terminate the episode. The behaviour policy is $\epsilon - greedy$ with $\epsilon = 0.1$. Value functions are tabular and initialized with 0.

Figure 11a demonstrates that, Q-learning starts with taking action left with probability surpassing 90%, revealing overestimation of $Q(A, left)$. In comparison, Double-Q learning does not overestimate $Q(A, left)$. However, VQ-learning is even more data efficient than Double Q-learning, while also avoiding any overestimation.

**Noisy value Updates.** Now for the same task example above, we remove the reward stochasticity and add Gaussian noises sampled from $\mathcal{N}(0, 0.01)$ to tabular-stored values to simulate noisy updates of function approximation techniques such as neural networks. As shown in Figure 11b, we observe similar results to those above. Furthermore, the advantage of VQ-learning over the other two methods is even more pronounced.

**Removing stochasticity.** Additionally, we investigate the convergence speed of all methods when the reward function and value function updates are deterministic rather than stochastic. In other words, the potential factors which can trigger overestimation of Q-learning are eliminated. As presented in Figure 11c, Q-learning does not overestimate the value function anymore and VQ-learning remains to be the most sample-efficient approach.

The rationale behind this is that, regardless of the method used, the number of execution times of the action with the highest value in state $B$ is strictly equal to or lower than the number of times state $B$ is visited. Therefore, in VQ-learning, when updating $Q_t(A, left)$ and computing the target value $r + \gamma V_t(B)$, $V_t(B)$ is likely to have already been updated by any actions executed from state $B$ before time step $t$. In contrast, the target value of $Q_t(A, left)$ in Q-learning depends the maximum $Q$ value for actions in state $B$, denoted as $\max_{a'} Q_t(B, a')$. Since the $Q$ values are initialized to zero, $\max_{a'} Q_t(B, a')$ will stay at zero rather than becoming negative until all actions are taken at least once in state $B$.

Thus, VQ-learning can identify the optimal policy faster than Q-learning. Double Q-learning improves sample efficiency compared to Q-learning by utilizing two independent Q functions, but it is still outperformed by

VQ-learning. The observation presented in Figure 11c can be further magnified in the fourth variant task by increasing the number of available actions for state $B$ up to 100, as shown in Figure 11d.

In addition, to more closely simulate the settings of the deep versions of the methods evaluated in Figure 2, we also apply replay buffers and batch learning. Similar results as in Figure 2 are observed in Figure 11 (Appendix A.2). To broaden and enhance the validations from Figure 2, we also devise a 2-room grid world (Figure 12) inspired by the MDP design in Figure 1. This 2-room grid world presents a more challenging version of the task shown in Figure 1, owing to its expanded state space. Appendix A.3 discusses experimental results for the 2-room grid world, demonstrating the superiority of VQ-learning over Q-learning and Double Q-leaning at learning the optimal policy faster. Additionally Appendix A.3 also provides the empirical validation for the fourth condition outlined in Theorem 1. Similar observation for DVQN is provided in Appendix A.5.

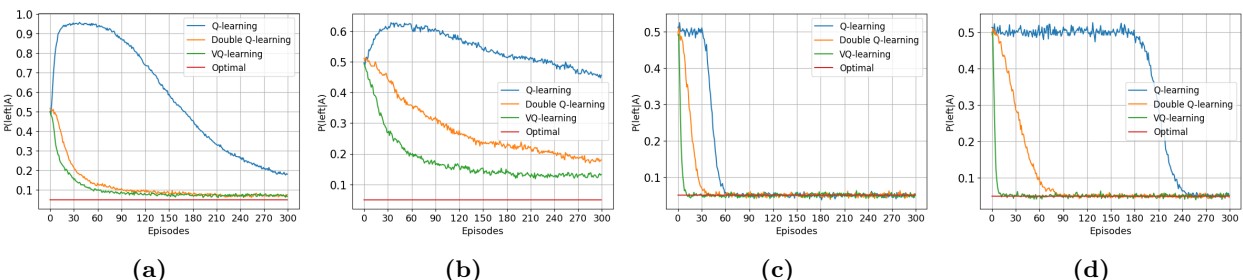

**Figure 2:** Probability of taking *left* from state $A$ against number of timesteps. The results are averages across 3000 repetitions. In the setting of **(a)**, actions from $B$ cause rewards sampled from Gaussian distribution $\mathcal{N}(-0.1, 1)$. In the setting of **(b)**, tabular values are added with independent Gaussain noises (sampled from $\mathcal{N}(0, 0.01)$) after each episode. Figure **(c)** depicts evaluations when applying deterministic reward function and value updates. The setting of **(d)** is identical to that of **(c)**, except for an increase in the number of available actions in state $B$ from 20 to 100.

## 4  Deep VQ-Networks

To overcome the overestimation problem in Deep Q-learning (DQN) without resorting to underestimation (Hasselt, 2010; Van Hasselt et al., 2016), we develop Deep VQ-Networks (DVQN) based on VQ-learning. We present the pseudo code for DVQN in Algorithm 2. A schematic diagram of our approach with a brief description of the architecture is provided in Figure 3.

In particular, along with the original Q function $Q_\phi(s, a)$, we learn additionally a state value function $V_\theta(s)$ according to temporal difference and use it to regulate the estimation of $Q_\phi(s, a)$. To elaborate, in lines 11-14 of Algorithm 2, $V_\theta(s)$ is updated according to its own TD-loss, no max operator is involved, $V_\theta$ is therefore not vulnerable to overestimation. In lines 15-21 of Algorithm 2, we construct the loss with respect to $\phi$ by combining two terms: the original Q-learning loss and the VQ-learning loss, weighted by the factor $\alpha$. $\mathcal{L}_{VQ}$ in form of knowledge distillation loss prevents $Q_\phi(s, a)$ from deviating far from $y_v(s', r)$ (line 17, Algorithm 2).

In contrast to employing only the VQ-learning loss as in tabular VQ-learning, our empirical findings suggest that combining the VQ-learning loss with the original Q-learning loss (adjusted by $\alpha$) yields superior performance in deep settings.

More specifically, when $y_q(s', r)$ happens to be overestimated, $y_v(s', r)$ is more likely to estimate a rational value for state $s$, which is more promising for $Q_\phi(s, a)$ to approach in this case. On the other hand, when the $y_q(s', r)$ is underestimated due to noisy updates of the neural networks, $y_v(s', r)$ plays a role of cross validation to regulate the update to $Q_\phi$, since $V_\theta$ is updated independent from $Q_\phi$. Moreover, when the target $y_q(s', r)$ for $Q_\phi(s, a)$ is rationally estimated without biased estimation, $y_v(s', r)$ will not deviate too much from $y_q(s', r)$ since the training data is shared between updates of $V_\theta$ and $Q_\phi$.

To clarify more regarding $\mathcal{L}_{VQ}$, it only updates parameters of $Q_\phi(s, a)$, thereby preventing the parameters of $V_\theta(s)$ from being influenced by potential overestimation (or underestimation) of Q values. The form of $\mathcal{L}_{VQ}$ is linked to knowledge distillation loss (Hinton et al., 2015) where $Q_\phi(s, a)$ is allowed to distill knowledge from $V_\theta(s)$ by approaching the estimation of $y_v(s', r)$, but not the other way around. The effect of this knowledge distillation can be adjusted by the hyperparameter $\alpha$.

As discussed in section 3, tabular VQ learning not only effectively addresses overestimation issues but also possesses an inherent advantage in terms of sample efficiency when compared to Q-learning, which also applies to DVQN. To offer a rationale for the latter advantage, when DVQN updates $Q_\theta(s, a)$ at timestep $t$ using data $(s, a, s', r)$, transitions that are sampled before $t$ from $s'$ with any available action may have already contributed to the current value of $V_\theta(s')$, which is used to update the Q network. In DQN, the action with the highest Q-value in state $s'$ is executed less frequently than the number of times state $s'$ being visited, resulting in the TD-error that are less informative compared to the one of DVQN.

In practice, as shown in Figure 3, the Encoder (CNN feature extractor) is shared between two streams of networks: V net and Q net. Both streams have MLP (Multi-Layer Perceptron) modules to estimate state and action values respectively, the former has a single output and the latter has an output size equal to the size of the action space. SG indicates *stop gradient*. The additional computational cost of DVQN, in comparison with DQN, primarily stems from the MLP part (two-layer) of the V net. As an easy-to-implement extension of DQN, DVQN does not sacrifice sample efficiency since $V_\theta$ and $Q_\phi$ use the same data for updates.

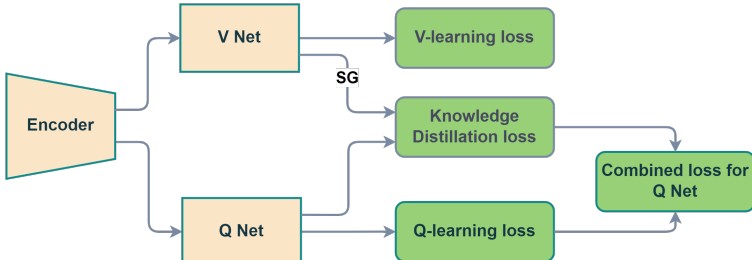

**Figure 3:** Architecture of DVQN. The encoder is shared between the state value network (V Net) and the action value network (Q Net) to improve training efficiency. V Net is learned on its own temporal difference loss, Q Net is learned by minimizing the combined loss of the original Q-learning loss and the discrepancy between estimations of V Net and Q Net in form of Knowledge Distillation. In other words, Q Net can learn from V Net but not vice versa. SG denotes *stop gradient*.

## 4.1 Comparison with Dueling DQN

Dueling DQN is known as a technique to improve sample efficiency over DQN, more introduction can be found in section 2. Dueling DQN incorporates a network stream for the state value function and another one for the advantage value function, their outputs are combined to estimate Q values. On the other hand, DVQN learns a separate network alongside the original Q network to estimate state value function independently. This distinction also implies that, DVQN does not increase scale of differentiable weights in the networks as compared to Dueling DQN.

Due to the Dueling architecture, Dueling DQN needs to address the identifiability issue by applying a constraint such as zero mean to the advantage estimation before combining it with the state value estimation. In contrast, the stream of state value networks in DVQN is trained independently with its own TD-loss, it inherently avoids the identifiability issue and therefore does not require such a constraint which could sacrifice the learning of Q value estimation.

Moreover, Dueling DQN relies on an assumption that the value for each action does not need to be learned for many states where the choice of actions is task-irrelevant. In contrast, DVQN does not necessarily require this assumption.

---

**Algorithm 2** DVQN

---

1: Initialize the environment and replay buffer $\mathcal{D}$
2: Initialize $Q$ net parameters $\phi$, $V$ net parameters $\theta$
3: Initialize target networks $\hat{\phi} \leftarrow \phi, \hat{\theta} \leftarrow \theta$
4: Initialize weighting factor $\alpha$, replay period $n$, update target networks period $m$
5: **for** $t = 1$ to $T$ **do**
6:     Select action $a$ given state $s$ ($\epsilon - greedy$)
7:     Observe reward $r$ and next state $s'$
8:     Store transition $(s, a, s', r)$ to replay buffer $\mathcal{D}$
9:     **if** $t \bmod n$ **then**
10:         Sample a batch of transitions $B = \{(s, a, s', r)\}$ from $\mathcal{D}$
11:         *# Compute loss for $V_\theta$:*
12:         $\mathcal{L}_V = \frac{1}{|B|} \sum_{(s,a,s',r) \in B} \left( r + \gamma V_{\hat{\theta}}(s') - V_\theta(s) \right)^2$
13:         Update $V_\theta$ by one step of gradient descent:
14:         $\nabla_\theta \mathcal{L}_V$
15:         *# Compute loss for $Q_\phi$:*
16:         $y_q(s', r) = r + \gamma \max_{a'} Q_{\hat{\phi}}(s', a')$
17:         $y_v(s', r) = r + \gamma V_\theta(s')$
18:         $\mathcal{L}_Q = \frac{1}{|B|} \sum_{(s,a,s',r) \in B} \left( y_q(s', r) - Q_\phi(s, a) \right)^2$
19:         $\mathcal{L}_{VQ} = \frac{1}{|B|} \sum_{(s,a,s',r) \in B} \left( y_v(s', r) - Q_\phi(s, a) \right)^2$
20:         Update $Q_\phi$ by one step of gradient descent:
21:         $\nabla_\phi ((1 - \alpha) \mathcal{L}_Q + \alpha \mathcal{L}_{VQ})$
22:     **if** $t \bmod m$ **then**
23:         Update target networks:
24:         $\hat{\phi} \leftarrow \phi$
25:         $\hat{\theta} \leftarrow \theta$

---

### 4.2 Combining DVQN with Temporal Contrastive Learning

Based on the architecture of DVQN, we impose an additional temporal contrastive learning objective (TC) to the feature extraction (encoder) module of DVQN to investigate its effect on reward performance. The temporal contrastive objective is constructed such that states that are temporally close to each other share similar representations in a learned latent space, while those that are temporally far from each other have dissimilar representations. The temporal contrastive objective stems naturally from the underlying markov decision process (MDP), thereby serves as an ideal task-agnostic auxiliary objective. Besides, works like Xue et al. (2023); Waradpande et al. (2020); Stooke et al. (2021) also show that the use of the underlying structure of the MDP in learning state representations leads to improvements in convergence and reward performance of RL algorithms in several domains.

In particular, we utilize two consecutive states in one transition as an anchor-positive pair to learn a rich temporal-level representation. Each anchor takes the positive samples corresponding to other anchors in the same batch data as its negative samples. In practice, we apply InfoNCE loss (Oord et al., 2018) to optimize the discrimination objective:

$$\mathcal{L} = -\frac{1}{|\mathcal{B}|} \sum_{q_i \in \mathcal{B}} \log \frac{\exp\left(sim(q_i, k_i)\right)}{\sum_{k_j \in \mathcal{K}} \exp\left(sim(q_i, k_j)\right)} \tag{9}$$

where $sim(q_i, k_j)$ denotes the similarity between codings of an anchor and its positive sample respectively.

To facilitate a comparative analysis, we also apply the same objective to the encoder of DQN. The results indicate DVQN combined with TC achieves better performance, more details are refer to section 5.2.5. We also experimented using CURL (Laskin et al., 2020), however, it does not contribute to performance

improvement for both DVQN and DQN. More details about our temporal contrastive objective and CURL can be found in Appendix A.6 and Appendix A.7.

## 5 Experiments and Analysis

### 5.1 Experimental Setup

**Experimental domains** We conduct our experiments on Atari 100k (Kaiser et al., 2019) benchmark, which is a sample-constrained benchmark for algorithms dealing with high-dimensional observations (raw pixel) and discrete control. The agent is allowed to interact for 100k steps in total with the environment, roughly equivalent to 2 hours of gameplay by human. To investigate longer-term performance of DVQN and baseline approaches, we conduct additional experiments with one million training steps. The evaluations are carried out in five arbitrarily chosen Atari domains, namely Asterix, Pong, Riverraid, Breakout and Boxing.

**Baseline approaches** We compare DVQN (our method) with baseline approaches, including DQN (Mnih et al., 2015), DDQN (Van Hasselt et al., 2016), CDDQN (Fujimoto et al., 2018) Averaged DQN (Anschel et al., 2017) and Dueling DQN (Wang et al., 2016),

**Evaluation** Our experimental evaluation is divided into five parts: We **first** compare our method to the baseline approaches in terms of reward performance in Atari 100k setting. **Second**, we investigate the robustness and sample efficiency of DVQN when expanding the action spaces with redundant actions. **Third**, we extend the training horizon up to one million steps to evaluate long-term performance of all methods above. **Fourth**, we investigate the impact of the hyperparameter $\alpha$ in DVQN (line21, Algorithm 2). **Last**, we analyze the effect of employing an additional temporal contrastive objective for feature extraction (encoder) in DVQN. We also compare it when DQN is combined with the same objective.

Observations need to be augmented by random shift (DrQ) (Kostrikov et al., 2020) before feeding them into the networks. DrQ has been testified on Atari 100k setting to be a simple yet effective method to improve sample efficiency and robustness of DQN (Kostrikov et al., 2020). Instead of solely integrating DrQ into DQN, we apply DrQ to each method included in our evaluation to ensure fair performance comparisons.

Our experimental results on 100k setting are averaged across 20 random seeds along with confidence interval of a half standard deviation. The results for experiments with one million steps are illustrated as averages over 7 random seeds. Hyper-parameters for the experiments are detailed in Appendix A.4. We keep the common hyper-parameters shared across all methods same for fair comparisons.

### 5.2 Result Analysis

#### 5.2.1 Performance on 100k setting

In Figure 4, we assess the performance of the reward against time steps. Our method, DVQN, demonstrates superior performance compared to DQN, DDQN, CDDQN, Averaged DQN and Dueling DQN across all domains. To specify in more detail, DDQN exhibits a modest improvement in reward performance over DQN in three out of five domains. On the other hand, CDDQN only enhances reward performance in two domains, while displaying even lower reward performance in two others. In contrast, DVQN (our method) showcases a substantial boost in reward performance over DQN significantly and outperforms DDQN, CDDQN and Averaged DQN in all domains. This indicates that DVQN consistently regulates the Q-value estimation in a positive manner. In addition, Dueling DQN also achieves improved reward performance compared to DQN but remains inferior to DVQN in all five domains. Hence, Figure 4 reflects that DVQN can effectively regulate value estimation while simultaneously enhancing sample efficiency.

More over, we track the estimation of Q value for each method, as shown in Figure 5. Given the fact that DVQN demonstrates the best reward performance in our experiments, the baseline methods tend to produce a higher bias in Q estimation (either over or underestimation).

In Appendix A.5, we demonstrate that the discrepancy in estimations between the V network and Q network is empirically shown to remain close to zero, providing further evidence in support of the fourth condition outlined in Theorem 1.

Note that our empirical findings indicate that the performance of Averaged DQN is sensitive to the specific interval chosen for updating the target network. We therefore further explore different update intervals of target networks for Averaged DQN. Figure 6 demonstrate that there is no best choice of update interval across all domains and DVQN consistently outperformed Averaged DQN across all interval options in each domain. The update interval for DVQN and DQN remains unchanged as 1000 steps. In addition, for Averaged DQN, the number $K$ of averaged target networks is set to 10, which exhibits the best performance in Anschel et al. (2017).

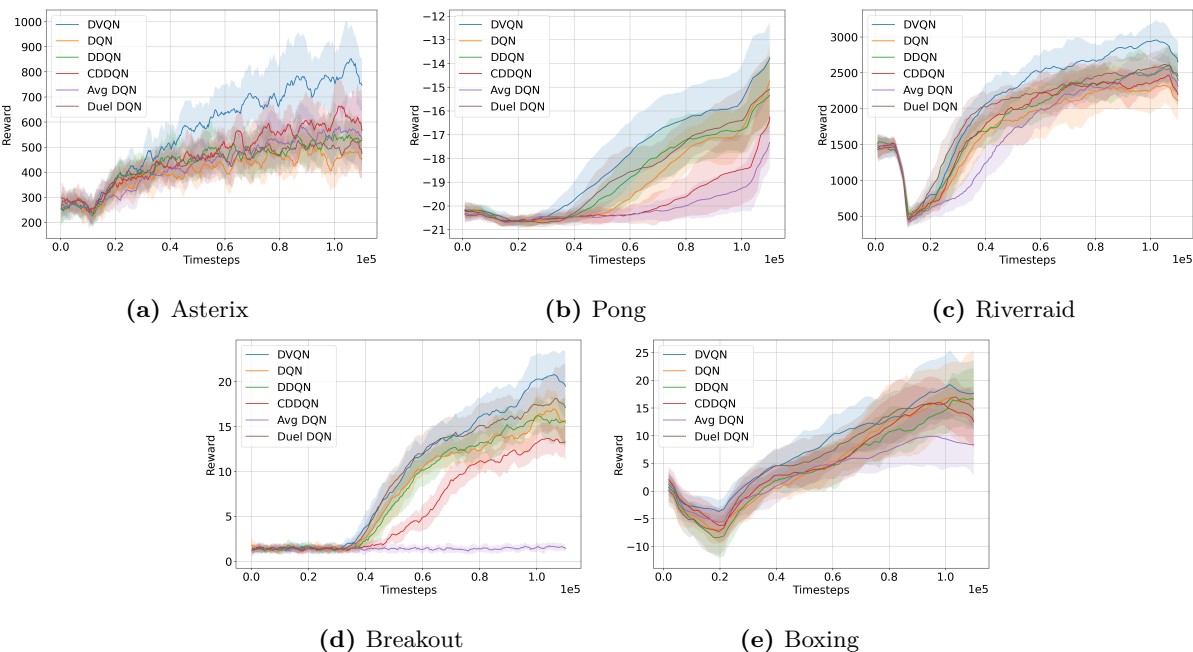

**Figure 4:** Reward performance against timesteps

### 5.2.2 Experiments with redundant actions

In order to delve deeper regarding robustness of DVQN, we expand the actions spaces of all domains with 10 additional no-operation actions, that makes each domain more challenging in terms of learning efficiency. As depicted in Figure 7, while DQN's performance declines across all five domains when utilizing the redundant action space instead of the original action space, DVQN with the redundant action space manages to maintain a comparable level of performance compared to DVQN with the original action space. The results reflect discussions in section 3.1 and 4.

### 5.2.3 Long-term performance

In order to investigate long-term performance of DVQN in comparison to the baseline methods, we also conduct experiments with one million training steps. As shown in Figure 8, DVQN maintains its best performance in three domains. In domain Breakout and Boxing, DVQN outperforms other baselines, with the exception of being surpassed by CDDQN during the latter stages of training. Nevertheless, it is noteworthy that DVQN exhibits a more noticeable advantage in reward performance during the earlier training phases.

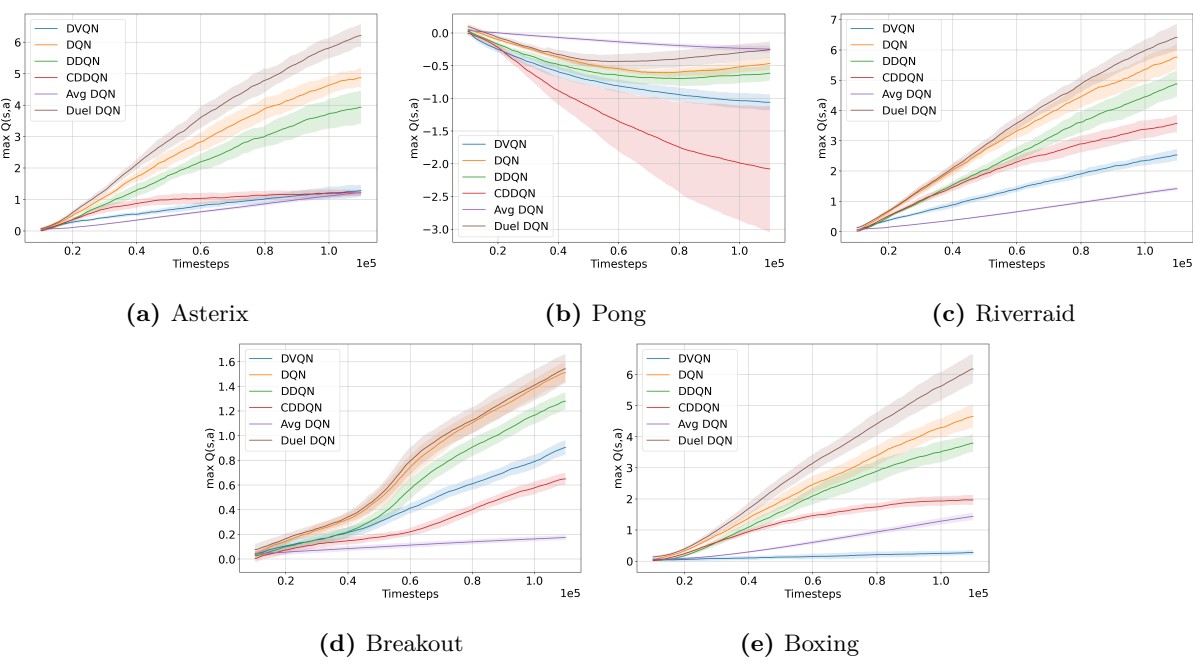

**(a)** Asterix        **(b)** Pong        **(c)** Riverraid

**(d)** Breakout        **(e)** Boxing

**Figure 5:** Max Q value estimation against timesteps

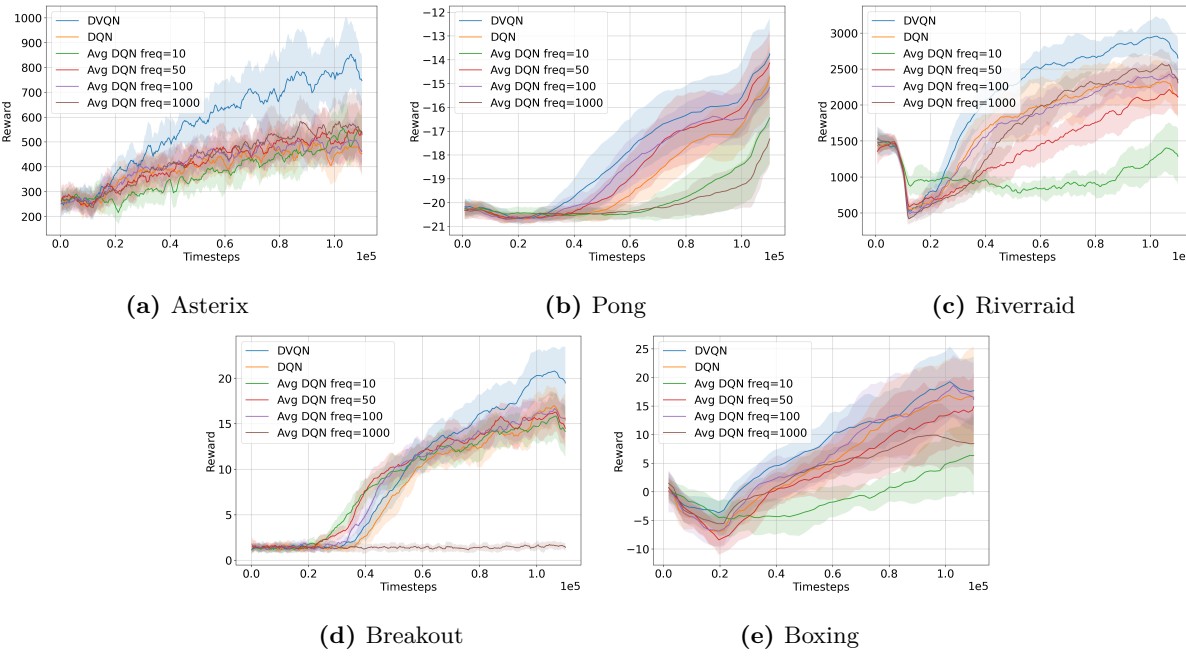

**(a)** Asterix        **(b)** Pong        **(c)** Riverraid

**(d)** Breakout        **(e)** Boxing

**Figure 6:** Reward performance of DVQN, DQN and Averaged DQN with different settings of update interval for target networks, e.g. freq=10 indicates the target network is updated every 10 steps

### 5.2.4   Effect of $\alpha$

In DVQN (Algorithm 2), the hyperparameter $\alpha$ adjusts the blend of the VQ-learning loss and the original Q-learning loss. By default, $\alpha$ is configured to 0.5. When $\alpha$ is adjusted to 1.0, DVQN exclusively relies on the VQ-learning loss, aligning with the tabular VQ-learning. With one million training steps, the effect of $\alpha$ is also explored. Our experimental results (Figure 9) indicate that DVQN with $\alpha = 0.5$ achieves equal or

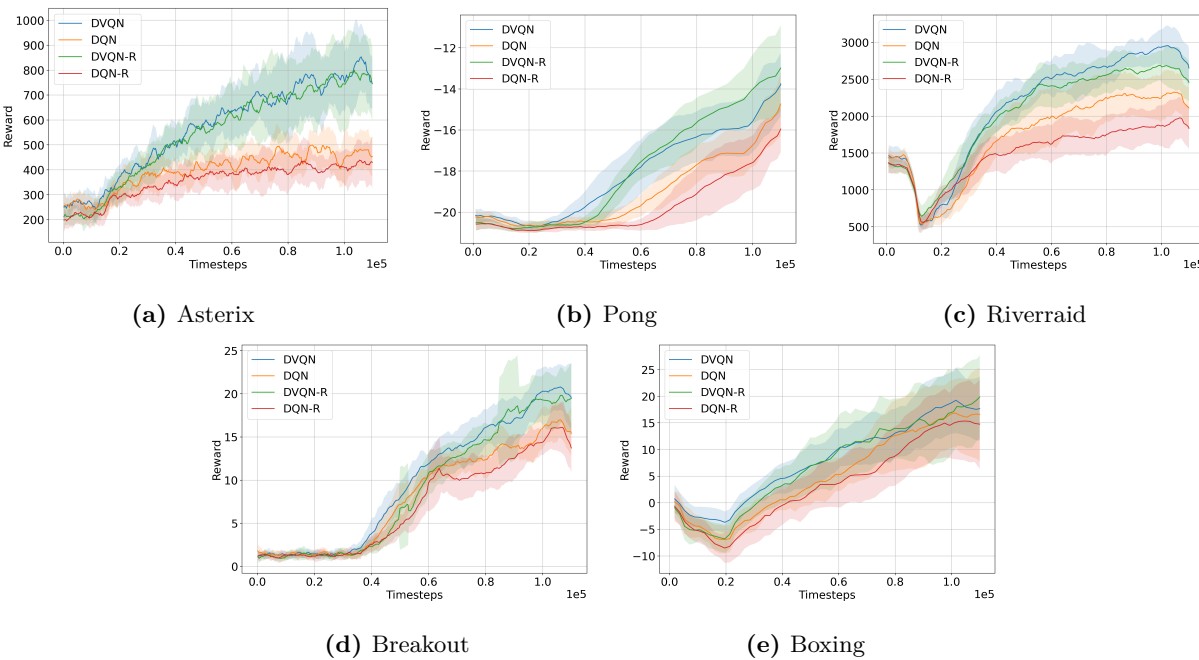

**(a)** Asterix

**(b)** Pong

**(c)** Riverraid

**(d)** Breakout

**(e)** Boxing

**Figure 7:** Reward performance of DVQN and DQN using expanded action space (adding 10 redundant no-operation actions, marked with suffix -R), respectively compared with DVQN and DQN using the original action space.

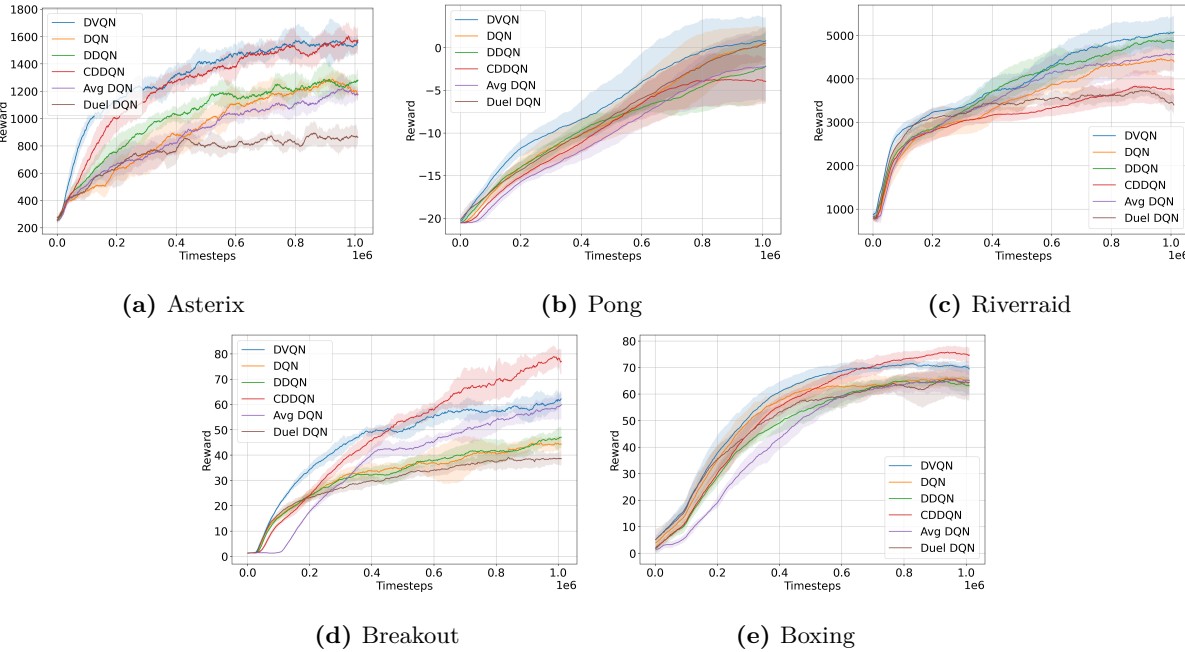

**(a)** Asterix

**(b)** Pong

**(c)** Riverraid

**(d)** Breakout

**(e)** Boxing

**Figure 8:** Reward performance with one million training steps.

better reward performance than $\alpha = 1.0$ in all five domains. The findings suggest that, despite the presence of potential overestimation induced by the original Q-learning loss, VQ-learning loss at the same time is able to effectively mitigate bias in Q-value estimation

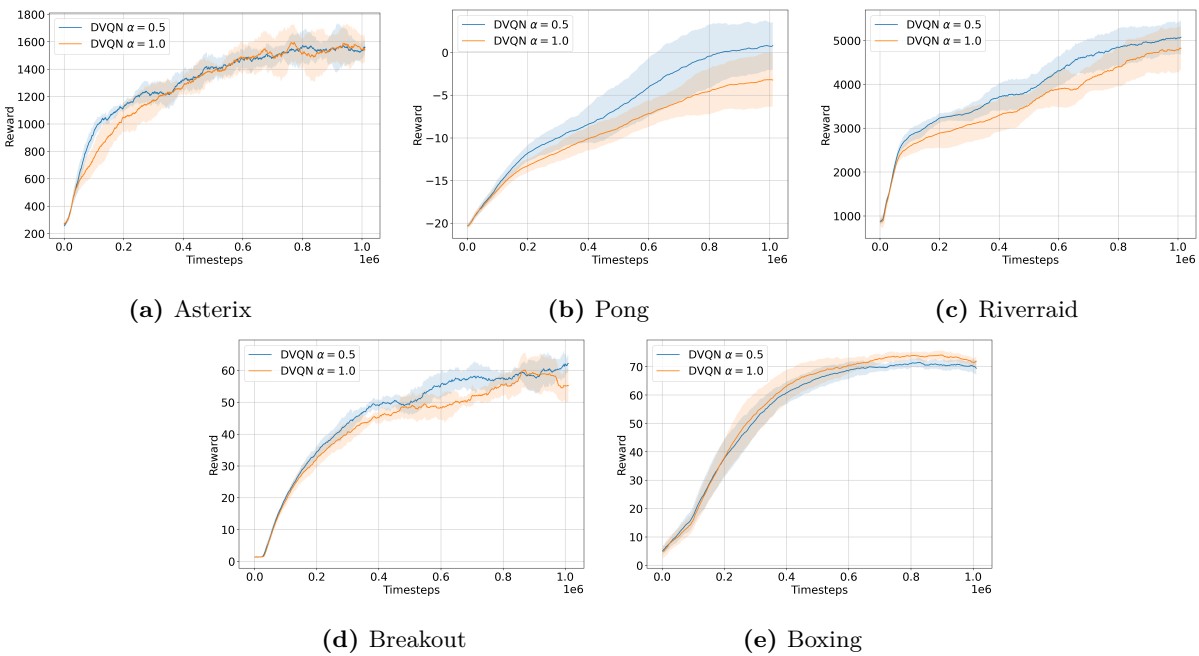

**(a)** Asterix        **(b)** Pong        **(c)** Riverraid

**(d)** Breakout        **(e)** Boxing

**Figure 9:** Reward performance with $\alpha$ equals to 0.5 and 1.0

### 5.2.5 Effect of temporal contrastive learning.

While DVQN helps alleviate the negative impact of network updates caused by biased value estimation errors, another goal is to investigate the impact on reward performance when an additional representation learning objective is introduced to DVQN. To be more precise, we employ DVQN and DQN as the backbone network architectures and incorporate an additional temporal contrastive learning objective (TC) into the feature extraction (encoder).

As shown in Figure 10, the combination of DVQN with TC leads to an improved reward performance in three domains compared to DVQN alone, while maintaining comparable reward performance in the other two domains. However, combining DQN with TC results in a notable decline in reward performance in Pong and Boxing, no improvement in Asterix. Although integrating TC with DQN enhances reward performance in Riverraid and Breakout, it still falls short of the performance achieved by DVQN combined with TC.

Based on the results presented, we conclude that DVQN serves as a better backbone network than DQN in harnessing the potential benefit of TC objective. This superiority can be attributed to our approach of regulating Q-function estimates, which minimizes detrimental network updates resulting from biased value estimations. DVQN, as a backbone network, facilitates the more efficient learning of temporal-level knowledge.

## 6 Conclusion

We introduce VQ-learning and its deep learning extension, DVQN, to address the issue of biased estimation of action values. In particular, our approach is based on learning an independent network to estimate state values which are then used to regulate the original Q-network in DQN. We provide empirical evidence showcasing that our method outperforms DQN, DDQN, CDDQN, Averaged DQN and Dueling DQN in terms of reward performance. Notably, our method is easy to implement, it comes with little increase in computational cost as compared to DQN and does not sacrifice data efficiency. Moreover, our experiments incorporating an additional temporal contrastive loss demonstrate that, using DVQN as the backbone network allows for greater potential benefits of the additional contrastive objective compared to using DQN. The idea of DVQN can be adapted in continuous control domains in the future work, since such an independent state value network can easily be incorporated into actor-critic settings. In comparison, Dueling DQN is only applicable

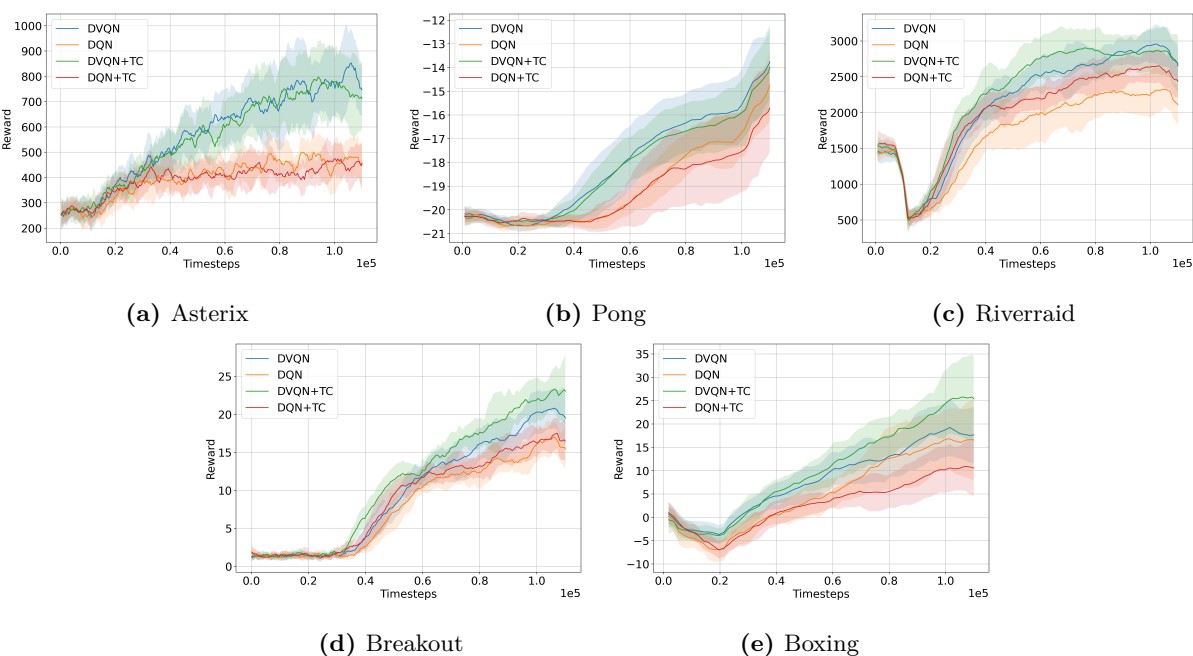

**(a)** Asterix  **(b)** Pong  **(c)** Riverraid

**(d)** Breakout  **(e)** Boxing

**Figure 10:** Reward performance of DVQN and DQN with temporal contrastive objective

in discrete control domains. Another future work can be to consider adaptive way of adjusting $\alpha$, since the optimal $\alpha$ may differ across domains.

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

# A    Appendix

## A.1    Convergence and Optimality of VQ-learning

To prove Theorem 1 we will need the following lemma by Singh et al. (2000), which is an extension of Theorem 1 from the work of Jaakkola et al. (1993).

**Lemma 1.** *Consider a stochastic process* $(\alpha_t, \delta_t, F_t)$, $t \geq 0$ *where* $\alpha_t, \delta_t, F_t : X \to \mathbb{R}^n$ *satisfy the equations:*

$$\Delta_{t+1}(x) = (1 - \alpha_t(x)) \Delta_t(x) + \alpha_t(x) F_t(x)$$

*where* $x_t \in X$ *and* $t = 0, 1, 2, ....$ *Let* $P_t$ *be a sequence of increasing* $\sigma - fields$ *such that* $\alpha_0$ *and* $\delta_0$ *are* $P_0$-*measurable and* $\alpha_t$, $\Delta_t$ *and* $F_t$ *are* $P_t$-*measurable, t=1,2,....* $\delta_t$ *converges to zero w.p.1 when the following conditions are satisfied*

1. *the set* $X$ *is finite*

2. $0 \leq \alpha_t \leq 1, \sum_t \alpha_t(x) = \infty$ *and* $\sum_t \alpha_t^2(x) < \infty$ *w.p.1*

3. $\|\mathbb{E}\left[F_t(x) \mid \mathcal{F}_t\right]\|_W \leq \gamma \|\Delta_t\|_W + C_t$, *with* $\gamma \in [0, 1)$ *and* $C_t$ *converges to zero w.p.1*

4. $\mathrm{Var}\left[F_t(x) \mid \mathcal{F}_t\right] \leq K\left(1 + \|\Delta_t\|_W^2\right)$, *for* $K > 0$

**Theorem 1.** *Consider a finite ergodic MDP and apply a GLIE learning policy* $\pi^Q$ *(i.e. non-greedy actions according to Q function are chosen with vanishing probabilities). Assume that at time step t, action* $a_t$ *is chosen according to* $\pi^Q$, $Q = Q_t$. $V_t$ *and* $Q_t$ *are updated to* $V_{t+1}$ *and* $Q_{t+1}$ *as follows:*

$$V_{t+1}(s_t) = (1 - \alpha_t) V_t(s_t) + \alpha_t (r_t + \gamma V_t(s_{t+1})) \tag{10}$$
$$Q_{t+1}(s_t, a_t) = (1 - \alpha_t) Q_t(s_t, a_t) + \alpha_t (r_t + \gamma V_t(s_{t+1})) \tag{11}$$

$Q_t$ *converges w.p.1 to the optimal Q-function as long as:*

1. $Q$ *and* $V$ *values are stored in a lookup table.*

2. $\mathrm{Var}[r(s, a)] \leq \infty$

3. $0 \leq \alpha_t \leq 1, \quad \sum_t \alpha_t(x, a) = \infty, \quad \sum_t \alpha_t^2(x, a) < \infty$

4. $\lim_{t \to \infty} V_t(s) = \lim_{t \to \infty} Q_t(s, a_t)$, $a_t = \pi^{Q_t}(s)$, *where* $\pi^{Q_t}(s)$ *is the GLIE policy derived from* $Q_t$

*Proof.* Let $Q^*$ denote the target $Q$ values. By subtracting $Q^*(s_t, a_t)$ from both sides of equation 11 we obtain:

$$
\begin{aligned}
Q_{t+1}(s_t, a_t) - Q^*(s_t, a_t) &= (1 - \alpha) Q_t(s_t, a_t) + \alpha(r_t + \gamma V_t(s_{t+1})) - (1 - \alpha + \alpha) Q^*(s_t, a_t) \\
&= (1 - \alpha) Q_t((s_t, a_t) + \alpha(r_t + \gamma V_t(s_{t+1})) - (1 - \alpha) Q^*(s_t, a_t) - \alpha Q^*(s_t, a_t) \\
&= (1 - \alpha)\left[Q_t(s_t, a_t) - Q^*(s_t, a_t)\right] - \alpha\left[r_t + V_t(s_{t+1}) - Q^*(s_t, a_t)\right].
\end{aligned}
\tag{12}
$$

To use Lemma 1 we define $x_t = (s_t, a_t)$ and denote:

$$\Delta_t(s_t, a_t) := Q_t(s_t, a_t) - Q^*(s_t, a_t) \tag{13}$$
$$F_t(s_t, a_t) := r_t + \gamma V_t(s_{t+1}) - Q^*(s_t, a_t) \tag{14}$$

Adding and subtracting $\gamma Q_t(s_{t+1}, a_{t+1})$ from $F_t(s_t, a_t)$, we obtain:

$$F_t(s_t, a_t) = r_t + \gamma Q_t(s_{t+1}, a_{t+1}) - Q^*(s_t, a_t) + \gamma\left[V_t(s_{t+1}) - Q_t(s_{t+1}, a_{t+1})\right] \tag{15}$$

where $a_{t+1} = \pi^{Q_t}(s_{t+1})$, $\pi^{Q_t}$ is *GLIE* policy derived from $Q_t$. Let:

$$F_t(s_t, a_t) = F_t^{\text{Sarsa}} + C_t^{VQ} \tag{16}$$

where

$$F_t^{\text{Sarsa}} = r_t + \gamma Q_t(s_{t+1}, b) - Q^*(s_t, a_t) \tag{17}$$

$$C_t^{VQ} = \gamma \left[ V_t(s_{t+1}) - Q_t(s_{t+1}, a_{t+1}) \right] \tag{18}$$

$F_t^{\text{Sarsa}}$ would correspond to $F_t$ in Lemma 1 if the Q values were updated according to SARSA. To apply Lemma 1 we denote the $\sigma$-algebra generated by the random variables $\{s_t, \alpha_t, a_t, r_{t-1}, \ldots, s_1, \alpha_1, a_1, Q_0\}$ by $P_t$. Note that $Q_t, Q_{t-1}, \ldots, Q_0$ are $P_t$-measurable and, thus, both $\Delta_t$ and $F_t$ are $P_t$-measurable, satisfying the measurability conditions of Lemma 1.

According to Theorem 1 in Singh et al. (2000) we have that:

$$
\begin{aligned}
\|E\{F_t^{\text{Sarsa}}|P_t\}\| &= \|E\{F_t^{\text{Q}} + C_t|P_t\}\|_\infty \\
&= \|E\{F_t^{\text{Q}}|P_t\} + E\{C_t|P_t\}\|_\infty \\
&\leq \|E\{F_t^{\text{Q}}|P_t\}\|_\infty + \|E\{C_t|P_t\}\|_\infty \\
&\leq \|\Delta_t\|_\infty + \|E\{C_t|P_t\}\|_\infty
\end{aligned}
\tag{19}
$$

for all $t$, where $F_t^{\text{Q}} = r_t + \gamma \max_{b \in A} Q_t(s_{t+1}, b) - Q^*(s_t, a_t)$ and $C_t = \gamma[Q_t(s_{t+1}, a_{t+1}) - \max_{b \in A} Q_t(s_{t+1}, b)]$, $\|E\{C_t|P_t\}\|_\infty$ converges to zero w.p.q under the GLIE policy.

Then We have:

$$
\begin{aligned}
\|E\{F_t(s_t, a_t)|P_t\}\|_\infty &= \|E\{F_t^{\text{Sarsa}} + C_t^{VQ}|P_t\}\|_\infty \\
&= \|E\{F_t^{\text{Sarsa}}|P_t\} + E\{C_t^{VQ}|P_t\}\|_\infty \\
&\leq \|E\{F_t^{\text{Sarsa}}|P_t\}\|_\infty + \|E\{C_t^{VQ}|P_t\}\|_\infty \\
&\leq \|\Delta_t\|_\infty + \|E\{C_t|P_t\}\|_\infty + \|E\{C_t^{VQ}|P_t\}\|_\infty
\end{aligned}
\tag{20}
$$

To satisfy condition 3 of Lemma 1, we are now left with showing $\|E\{C_t^{VQ}|P_t\}\|_\infty$ converges to zero w.p.1, which is satisfied by condition 4 in Theorem 1. In section A.3 and A.5, we demonstrate the support to the condition 4 by empirical results.

Condition 4 in Lemma 1 can be established from the similar property of $F_t^{\text{Sarsa}}$.

$\square$

## A.2   VQ-learning with replay buffer

We also apply replay buffer to all tabular methods investigated in section 3. Compared to the results in Figure 2, similar results (Figure 11) are observed.

## A.3   Evaluation in 2-room grid world

We also devise a 2-room grid world to evaluate VQ-learning against Q-learning and Double Q-learning, as shown in Figure 12. The grid world has two rooms separated by a wall in the middle while being connected by a door in the wall. The red cell denotes the starting state and overlaps the door. The two green cells denote goals, the goal state ($Goal\#1$) located in the lower room causes a reward sampled from $\mathcal{N}(-0.1, 1)$. The other goal state ($Goal\#2$) in the upper room causes a reward of 0.1. Therefore, the optimal policy is to reach $Goal\#2$. An ordinary step causes a reward of 0. This 2-room grid world can be be viewed as an intensified version of the toy example in section 3.1.

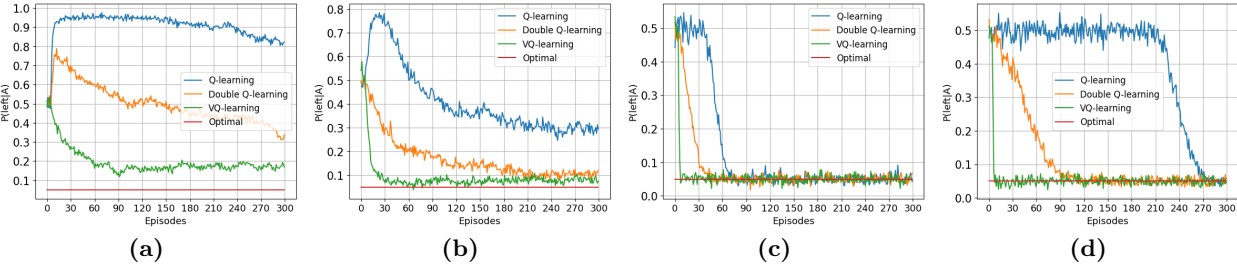

(a)  (b)  (c)  (d)

**Figure 11:** Results of applying a replay buffer of size 100 to VQ-learning, Double Q-earning and Q-learning, with batch size of 8 for training. The configurations of the four task variants, along with their settings, remain consistent when compared to the tabular VQ learning discussed in the main paper.

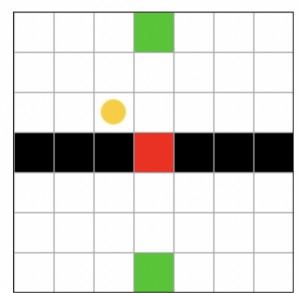

**Figure 12:** 2-room grid world

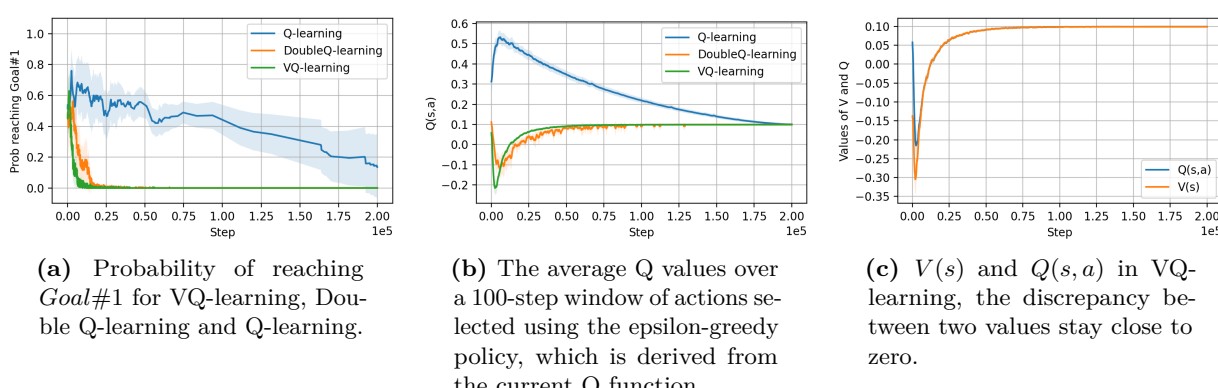

**(a)** Probability of reaching *Goal#1* for VQ-learning, Double Q-learning and Q-learning.

**(b)** The average Q values over a 100-step window of actions selected using the epsilon-greedy policy, which is derived from the current Q function.

**(c)** $V(s)$ and $Q(s, a)$ in VQ-learning, the discrepancy between two values stay close to zero.

**Figure 13:** Evaluation in 2-room grid world

The learning horizon of 200000, the learning rate of 0.1 and the exploration rate that exponentially decaying from 0.5 to zero are shared across all three methods. The results are averaged across 20 repetitions.

Figure 13a demonstrates that, Q-learning starts with probability of reaching *Goal#1* exceeding 50%. While Double Q-learning learns the optimal policy faster than Q-learning, VQ-learning outperforms both Q-learning and Double Q-learning. Figure 13b provides another perspective in terms of Q values: compared with VQ-learning, Q-learning begins by overestimating Q values higher than 0.5, Double Q-learning avoids overestimation and instead underestimates Q-values, slightly lower than VQ-learning.

Figure 13c illustrates that the discrepancy between $V_t(s_t)$ and $Q_t(s_t, a_t)$ is close to zero in VQ-learning, providing empirical support for the fourth condition in Theorem 1. Action $a_t$ is chosen given state $s_t$ according to the epsilon-greedy policy based on $Q_t$. With DVQN, similar observation can be found in A.5.

### A.4 Experimental settings and implementation details for DVQN

In section 5, we compare our method with baseline methods in five Atari game domains, namely Asterix, Breakout, Riverraid, Pong, and Boxing. All five domains share the same setting (as below) for preprocessing observations.

- Number of no-operation actions on reset: 30

- Frameskip (Number of frames skipped between steps): 4

- Framestack: 4

- Max-pooling (Pooling over the most recent two observations from the frame skips): True

- Resize: 84 x 84

- Grey scale observation: True

- Normalization: [0, 1)

Additionally, before feeding into the networks, the preprocessed observations need to be augmented by random shift (Kostrikov et al., 2020), which has been testified to be a simple yet effective method to improve sample efficiency and robustness of deep reinforcement learning algorithms. Furthermore, sizes of action space for each domains are listed in table 1.

**Table 1:** Action space sizes for each domain

| Domain | Action Space Size |
|---|---|
| Asterix | 9 |
| Pong | 6 |
| Breakout | 4 |
| Boxing | 18 |
| Riverraid | 18 |

All methods involved in our evaluation share the common deep reinforcement learning related hyperparameters, which are listed below:

- Number of warm-up steps with random actions before start of learning: 10000

- Number of total training steps: 100000

- Replay buffer size: 100000

- Batch size: 128

- $\gamma$: 0.99

- Exploration rate $\epsilon$: 0.1

- Learning rate: 0.0001

- Frequency of updating target networks: 1000 steps

- Rate of updating target networks $\tau$: 1

- Training frequency: 1

- Clip reward: $[-1, 1]$

- Clip gradient: $[-1, 1]$

- Optimizer: RMSprop

In DVQN, the weighting factor $\alpha$ is assigned a value of 0.5 to generally achieve a balance between the original Q-learning loss and VQ-learning loss. As for temporal contrastive learning, the momentum network is updated with a frequency of 1 and a rate of 0.001. Additionally, learning rate of 0.0001 is assigned for contrastive learning.

More details regarding the implementations of DVQN and other methods involved can be found in publicly released source code: `https://anonymous.4open.science/r/DVQN-046F`. We run our experiments using a single CPU core of AMD EPYC 7742 and a single GPU of NVIDIA DGX A100.

## A.5 Discrepancy between V net and Q net of DVQN

As illustrated in Figure 14, we track the difference between $V_t(s)$ and $Q_t(s,a)$, which empirically support the fourth condition outlined in Theorem 1. $(s,a)$ is batch data sampled from the replay buffer.

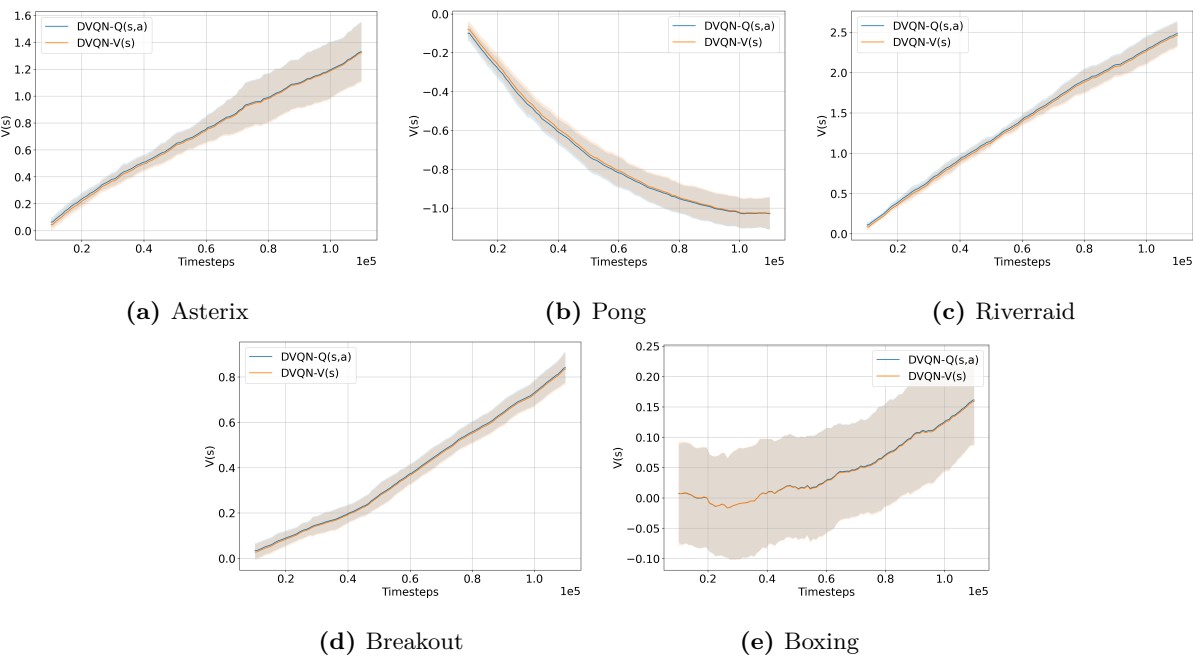

**Figure 14:** $V_t(s)$ and $Q_t(s,a)$ for DQN with replay buffer. The discrepancies remain close to zero.

## A.6 More details regarding temporal contrastive learning

We apply an additional temporal contrastive learning objective to the encoder part of both DVQN and DQN and evaluate its effect in terms of reward performance. Figure 15 illustrates our framework of temporal contrastive learning. It involves three learnable components: encoder $f$, projection head $g$ and transformation matrix $W$. The usage of projection head $g$ can help avoid features from being tightly clustered (Gupta et al., 2022), called feature collapse. More specifically, without the projection head, feature learning tends to be overwhelmed by the contrastive objective and ignore task-related knowledge. Besides, since the data in RL settings is generated and stored dynamically, to stabilize the temporal contrastive learning, we applied momentum encoder and momentum projection head (He et al., 2020; Laskin et al., 2020) to process $s_{t+1}$. $\hat{f}$ and $\hat{g}$ are synchronized with $f$ and $g$ respectively by a fix interval of time steps.

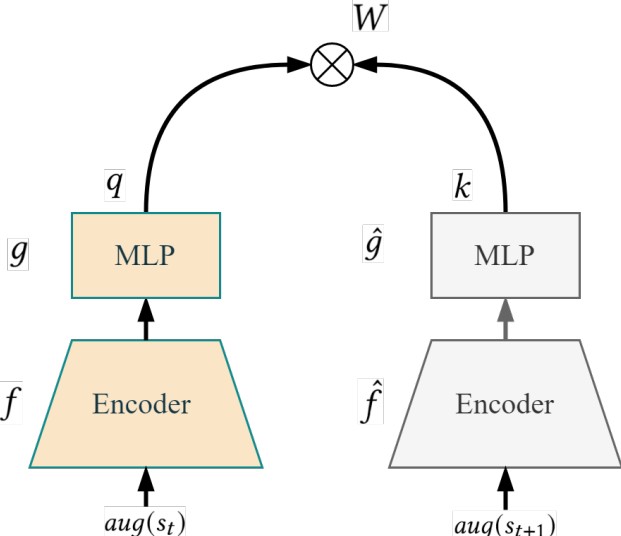

**Figure 15:** Network structure of temporal contrastive learning. $f$ and $g$ are encoder and projection head respectively, $\hat{f}$ and $\hat{g}$ are momentum counterparts. $s_t$ and $s_{t+1}$ are consecutive data sampled from the replay buffer, they are augmented by random shift (denoted by $aug$). Only encoder $f$ is shared with DVQN. $g$ is employed to prevent the output of the encoder from being overwhelmed by the discrimination objective and ignoring task-related knowledge.

The temporal contrastive objective we use shares similarity with ATC (Stooke et al., 2021) by sampling contrastive data pairs from temporally close states. ATC is proposed to decouple representation learning from deep reinforcement learning. In our setup, the convolutional encoder is trained by optimizing both temporal contrastive learning loss and reinforcement learning loss. Moreover, we use projection heads for both anchor and positive samples, whereas ATC only uses one projection head for the anchor sample.

### A.7 CURL as the additional objective

We also conduct experiments incorporating the contrastive objective introduced by CURL (Laskin et al., 2020) to DVQN and DQN. Compared with the temporal contrastive objective we employ, CURL involves using two random augmentations of a single state as a contrastive pair. We find in our experimental results (Figure 16) that training DVQN and DQN combined with the CURL loss results in equal or declined reward performance in all five domains, compared to using backbone networks alone. Even though DVQN is superior to DQN when combined with CURL in reward performance.

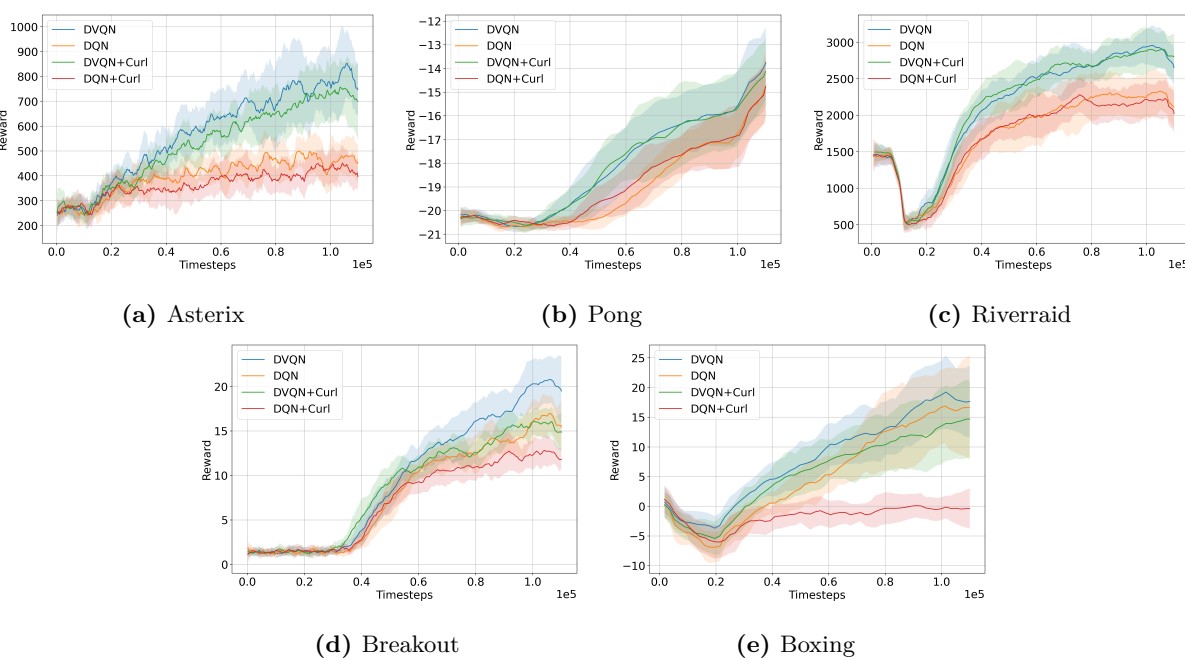

**(a)** Asterix        **(b)** Pong        **(c)** Riverraid

**(d)** Breakout        **(e)** Boxing

**Figure 16:** Reward performance against time steps of DVQN and DQN combined with CURL loss

