# OpenReview forum: "VQ-learning: Towards Unbiased Action Value Estimation in Reinforcement Learning"
_TMLR — Rejected by TMLR_

### Review · Reviewer_1HpF · 2024-06-02

**Summary Of Contributions:**

This paper proposes a new temporal difference learning algorithm called VQ-learning that simultaneously learns both a V-function and a Q-function. Compared to Q-learning and its variant, VQ-learning addresses the overestimation issue because it learns V and Q independently without the use of the max operator. Based on this algorithm the paper also presents the DQN analogue of Q-learning called DVQN.

**Audience:**

Yes

**Broader Impact Concerns:**

Not provided and not applicable.

**Claims And Evidence:**

No

**Requested Changes:**

- How is the proposed Algorithm 1 different from or related to SARSA?  SARSA and expected SARSA [see Barto & Sutton RL book Chapter 6.4] are also TD learning algorithms for control (learning the optimal policy) that do not use the max operator. I think omitting these as baselines is not justified, they should at minimum be included for Sec 2 and Sec 3.1.
- Alg 2 L16-L21: why not do the same as clipped DQN (similar to Eqn 6) instead of combining $y_q$ and $y_v$ with $\alpha$?
- How do you set $\alpha$? if alpha is 0 then does it become equivalent to regular DQN with target net?
- Page 7: "Q Net can learn from V Net but not vice versa" - V net learns from sample generated from Q net esp greedy
- Sec 4.2 describes the temporal contrastive learning objective, but it seems disconnected to the proposed method. Does the proposed method rely on temporal contrastive learning to work? If not, it should not be included in Sec 4 as part of the methods, and instead should be in Sec 5 experiments as implementation detail or a specific scenario of comparison.
  - Eqn 9 whats B and K?

- Sec 5.2.1 - Fig 5 I don't think you can claim other baselines are over/under-estimating but the proposed method is not over/under-estimating, since there is no justification that the proposed method leads to the true Q-values.  Is there a way to get the true Q values?
- Fig 6 how sensitive is the proposed method sensitive to target net update frequency? Here you only show how average DQN is sensitive to the update frequency, but the proposed method can have the same issue.
- Sec 5.2.4 "VQ-learning loss at the same time is able to effectively mitigate bias in Q-value estimation" if you believe the presence of the VQ-learning loss is able to "mitigate" the bias, then why isn't VQ loss alone the best (when alpha is 1)? Why is 0.5 the best? Have you tested other other values of alpha between 0 and 1.
- Fig 10 the description of results is not very accurate. "the combination of DVQN with TC leads to an improved reward performance in three domains compared to DVQN alone, while maintaining comparable reward performance in the other two domains." I think (a)(b)(c) are very similar performance, only (d)(e) the two domains improved. These improvements are also very small since the error bars (half std) are overlapping. Also DQN benefited more from TD in (c) Riverraid. Therefore I'm not convinced by the claim that DVQN is a better backbone based on this comparison result.

Minor issues
- Sec 3.1 on page 5-6 figure references should be fig 2 instead of fig 11

**Strengths And Weaknesses:**

- Exposition is clear and logical
- Provided good background information on related Q-learning based approaches

---

> ### Author Response · Authors · 2024-06-13
>
> We appreciate the detailed and insightful comments and we hope the concerns can be addressed as follows:
>
> **Q1:** How is the proposed Algorithm 1 different from or related to SARSA and Expected SARSA?
>
> **A1:** When applying epsilon-greedy policy, both SARSA and expected SARSA actually also use the max operator to compute the TD-target.
>
> In SARSA:
> TD-target is computed: $r(s,a) + \gamma Q(s’, \pi(s’))$, where $\pi(s’)=argmax(Q(s’))$ with probability of $1-\epsilon$.
>
> In Expected SARSA: $Q_{target} = r(s,a) + \gamma (\max Q(s’,a’) (1-\epsilon+\epsilon/k) + [\sum Q(s,’ a’) - \max Q(s’,a’)] \epsilon/k)$, $k$ is the number of actions.
>
> As long as the computation of the TD-target value involves max operator, even stochastically, the learning would be under the exposure of overestimation bias. In contrast, VQ-learning can avoid this since $V(s’)$ is the only estimated value used for computing the TD-target.
> Additionally, when computing the TD-target, Expected SARSA explicitly computes expectation of Q values over the next state, that requires $k$ times of table look-ups, whereas VQ learning only needs one time of table look-up $(V(s’))$. We will augment Sec 3.1 by comparing VQ-learning with SARSA and Expected SARSA as soon as possible.
>
> **Q2:** Alg 2 L16-L21: why not do the same as clipped DQN (similar to Eqn 6) instead of combining $y_q$ and $y_v$ with $\alpha$?
>
> **A2:** This is possible in principle, but the resulting inclusion of the minimum operator could then cause underestimation.
>
> **Q3:** How do you set $\alpha$? if alpha is 0 then does it become equivalent to regular DQN with target net?
>
> **A3:**  Yes, if \alpha is 0, then DVQN degenerates to DQN. In our preliminary experiments, we empirically compare  $\alpha=0.5$ and $\alpha = 1.0$ for DVQN. In Sec 5.2.4, we find setting $\alpha$ to 0.5 achieves better performance, so that the VQ loss is capable enough to regulate the Q values from biased estimation.
>
> **Q4:** Page 7: "Q Net can learn from V Net but not vice versa" - V net learns from sample generated from Q net esp greedy
>
> **A4:** You are right, updates for V net use data collected by Q net. We meant to say that *for each step of update*, VQ loss (Alg1, L19) involves estimation of V net, but only the parameters of Q net will be updated by applying stop gradient for V net. In contrast, V loss (Alg1, L12) does not involve Q net estimation, relying solely on bootstrapping the V net itself to compute TD error. We will clarify this in the paper.
> Even for a sample $(s,a,s’,r)$ that is collected due to the overestimated Q values, its target value is always computed without using the max operator, meaning the update to $Q(s, a)$ won’t be afflicted by overestimation.
>
> **Q5:** Does the proposed method rely on temporal contrastive learning to work? If not, it should not be included in Sec 4 as part of the methods.
> - Eqn 9 whats B and K?
>
> **A5:** VQ learning does not rely on additional temporal contrastive learning to work. Given the fact that there is a spectrum of methods proposing auxiliary representation learning objectives to improve learning efficiency of Deep RL. We want to additionally underscore that DVQN can also be an ideal choice of backbone network for auxiliary objectives. We agree that it is reasonable to move the corresponding part to Sec 5 and will do so.
> - B is batch size, K is the set of positive and negative samples for one anchor sample
>
> **Q6:** Is there a way to get the true Q values?
>
> **A6:** We are currently running experiments to compute additionally true Q values (as computed in the paper of Deep Double Q-learning) for VQ-learning. We will update the paper with new results as soon as possible.
>
> **Q7:** Fig 6 how sensitive is the proposed method sensitive to target net update frequency? Here you only show how average DQN is sensitive to the update frequency, but the proposed method can have the same issue.
>
> **A7:**  When we adjust the target net update frequency to different values, we notice that all methods maintain a stable relative performance, with the exception of Average DQN.  In order to not give DVQN an unfair advantage, we therefore tried different target net update frequencies exclusively for Averaged DQN to release a better performance in each domain. For other methods, we just set it to a common choice as 1000 for this hyperparameter.
>
> **Q8:** Why is 0.5 the best? Have you tested other other values of alpha between 0 and 1.
>
> **A8:** We empirically evaluated $\alpha=1.0$ and $\alpha=0.5$ in DVQN,  $\alpha = 0.5$ (mixing VQ loss and Q loss) results in better performance. We also tested other values between 0 and 1 and did not observe significant difference compare to 0.5.
>
> **Q9:** Fig 10 the description of results is not very accurate.  Error bars (half std) are overlapping in some domins.
>
> **A9:** Overlapping Variance intervals do not necessarily mean that the results are not statistically significant, as an alternative we will present standard error intervals for Fig 10.

---

> > ### Author Response · Authors · 2024-06-19
> >
> > Update to **A9**:
> >
> > In addition to the new findings mentioned in the official comment on the top, we have also replotted Fig. 10, substituting the variance with the standard error of the mean for the intervals. This change aims to better illustrate the stochastic significance of the mean performance.

---

### Review · Reviewer_yAoK · 2024-06-06

**Summary Of Contributions:**

This submission proposed VQ-learning, by simultaneously updating the value function and the state-action value function in the algorithm. As claimed by the authors, the proposed method avoids the use of the max operator, and can better handle both over- and under estimation for Q-values. The authors further extended it to the settings with deep networks by proposing the DVQN algorithm. The experimental results demonstrated the superior performance when compared against other methods.

**Audience:**

Yes

**Claims And Evidence:**

Yes

**Requested Changes:**

1. The condition 4 in Theorem 1 is a bit too strong. The authors need to show $\lim_{t \rightarrow \infty} V_t (s) = \lim_{t \rightarrow \infty Q_t (s, a_t)$, instead of assuming it to be true.
2. The benefits of temporal contrastive learning are a bit confusing: Why not use it as your main algorithm to compare against other methods since Figure 4? Furthermore, the setup for the positive (and negative) pairs may be problematic: It's quite possible that the two images could still be very similar to each other, even though they correspond to inadjacent states.
3. Some quite relevant papers on reducing overestimation bias in DQNs need to be discussed, e.g.,
* Song, Z., Parr, R. and Carin, L., 2019, May. Revisiting the softmax bellman operator: New benefits and new perspective. In International conference on machine learning (pp. 5916-5925). PMLR.
* Ren, Z., Zhu, G., Hu, H., Han, B., Chen, J. and Zhang, C., 2021. On the estimation bias in double q-learning. Advances in Neural Information Processing Systems, 34, pp.10246-10259.
4. Could you try to test the proposed method on more games, since you used Atari 100k?
5. Could you elaborate more on why VQ-learning can alleviate the issue of the underestimation bias?
6. The connection with the distillation loss (Hinton et al., 2015) at the top of Page 7 lacks rigor. Please share some details/derivations how value function can distill knowledge to the Q-function.

**Strengths And Weaknesses:**

Strengths:
1. The proposed method, VQ-learning, looks interesting, and the authors presents it in a mostly comprehensible manner.
2. The extension towards DVQN is reasonable and the algorithm shows some promise during empirical evaluation.

Weakness:
1. Given that a greedy policy corresponds to a fixed-point for the Bellman equation, it's unclear what the optimal policy should be in VQ-learning. Even though the authors removed the maximum operator, they still used it when choosing the action for the next step, i.e., L6 in Algorithm 1.
2. It's unclear to me why the proposed method can solve the issue of underestimation for Q-value, after removing the maximum operator.
3. The improvement from DVQN is a bit difficult to justify on the Atari games: The gap in Figure 4 is mostly obvious on Asterix and Riverraid, and 5 games seem limited given the use of Atari 100k.

---

> ### Author Response · Authors · 2024-06-13
>
> We appreciate the detailed and insightful comments and we hope the concerns can be addressed as follows:
>
> **Weakness**:
>
> **Q1:** Even though the authors removed the maximum operator, they still used it when choosing the action for the next step, i.e., L6 in Algorithm 1.
>
> **A1:** Yes, choosing the action still involves max operator. However, even for a sample $(s,a,s’,r)$ that is collected due to the overestimated Q values, its target value is always computed without using the max operator, meaning the update to $Q(s, a)$ won’t be afflicted by overestimation.
>
> **Q2:** It's unclear to me why the proposed method can solve the issue of underestimation for Q-value, after removing the maximum operator.
>
> **A2:** See A8
>
> **Q3:** The improvement from DVQN is a bit difficult to justify on the Atari games: The gap in Figure 4 is mostly obvious on Asterix and Riverraid, and 5 games seem limited given the use of Atari 100k.
>
> **A3:** In Figure4, we observe that DVQN also outperforms the baselines by a significant margin in Pong. And in Breakout,  DVQN converges to significantly higher reward than the baselines.
>
> **Requested Changes**:
>
> **Q4:** The condition 4 in Theorem 1 is a bit too strong. The authors need to show $\lim_{t \rightarrow \infty} V_t (s) = \lim_{t \rightarrow \infty Q_t (s, a_t)$, instead of assuming it to be true.
>
> **A4:** We acknowledge the limitation of our theoretical analysis, and unfortunately so far we were not able to find a complete proof of condition 4. However, we believe that the empirical evidence provided on condition 4 gives a sufficiently strong backing of our claim. Specifically, in Figure 13(c) and Figure 14, we measure the discrepancy between $V_t (s)$ and $Q_t (s, a_t)$ for VQ-learning and DVQN respectively.
>
> **Q5:** Why not use it as your main algorithm to compare against other methods since Figure 4?
> It's quite possible that the two images could still be very similar to each other, even though they correspond to inadjacent states.
>
> **A5:** We primarily aim to compare DVQN with the original configurations of baseline methods that do not incorporate additional representation learning objectives. Furthermore, considering the numerous studies that employ auxiliary representation learning objectives to enhance deep RL but often lead to fragile performance, we believe it is valuable to examine the performance of DVQN as a backbone network when integrated with a temporal contrastive learning objective.
>
> Two nonadjacent states might be visually similar to each other but could have significantly different value functions. On the other hand, two states that are adjacent in multiple samples would have similar value functions independent of how similar or dissimilar they are visually. And the goal of introducing an additional contrastive objective is to embed states with similar value functions closer. We, therefore, designed the technique so that temporally close states are treated as positive pairs as they would have similar value functions.
>
> **Q6:** Some quite relevant papers on reducing overestimation bias in DQNs need to be discussed
>
> **A6:** We thank the reviewer for the pointers and will include them in the discussion of the related work.
>
> **Q7:** Could you try to test the proposed method on more games, since you used Atari 100k?
>
> **A7:** We believe that evaluating five Atari games provides a meaningful assessment. Similarly, the previous works listed below have demonstrated significant results using a comparable number of Atari games. We agree that experiments with more Atari games can further augment the proposed work though.
> - Chen, Lili, et al. "Decision transformer: Reinforcement learning via sequence modeling."
> - Song, Zhao, Ron Parr, and Lawrence Carin. "Revisiting the softmax bellman operator: New benefits and new perspective."
>
> **Q8:** Could you elaborate more on why VQ-learning can alleviate the issue of the underestimation bias?
>
> **A8:** The source of underestimation can be the usage of minimum operator or the explicit bias built within DoubleQ-learning, both of them are absent in VQ-learning. We will also present the true Q values for a grid world domain.
>
> **Q9:** Please share some details/derivations how value function can distill knowledge to the Q-function.
>
> **A9:** Specifically, *for one update over a given transition*, although VQ loss (Alg1, L19) involves estimation of V net, only Q net will be updated. In contrast, V loss (Alg1, L12) does not involve Q net estimation, relying solely on bootstrapping the V net itself to compute the TD error.

---

### Review · Reviewer_dmbx · 2024-06-06

**Summary Of Contributions:**

The authors introduce VQ-learning, a novel approach that addresses the issue of action value overestimation inherent in Q-learning and Deep Q-learning in stochastic environments. They extend this method to Deep VQ Networks (DVQN) and demonstrate enhanced sample efficiency compared to Double Q-learning. Their evaluations on the Atari-100k benchmark reveal that DVQN consistently outperforms well-known RL algorithms.

**Audience:**

Yes

**Claims And Evidence:**

Yes

**Requested Changes:**

see  Strengths And Weaknesses part

**Strengths And Weaknesses:**

Strengths:

1. The VQ and Deep VQ Networks seem novel.

2. The algorithm demonstrates impressive numerical performance.

3. The paper is easy to follow.

Weaknesses:

1. The theoretical foundation in this paper is lacking. The authors do not offer a finite-time analysis or effectively discuss why their methods are superior. The concept of unbiased estimation in the algorithm is not adequately explained.

2. The motivation behind the paper is unclear. Although the authors reference previous work that employs additional samples to ensure unbiasedness, they do not effectively link this to their algorithm. The claim that VQ is a form of generalized policy iteration lacks presentation in the paper.

---

### Decision · Action_Editor_z6vF · 2024-07-18

**Recommendation:** Reject

**Comment:**

This paper proposes VQ-learning, which learns both the V- and Q-functions simultaneously. The algorithm is expected to mitigate over- and under-estimation of the Q-function, and can be extended to deep learning settings. Empirically, VQ-learning shows promising performance, compared to several baselines.

The topic is relevant and important. The proposed algorithm is supported by some theory and experiments. Overall, reviewers see potentials in this work, but agree more work is needed before the paper can be published. First, it is unclear how removing the “max” operation necessarily avoids both under- and over-estimation. A discussion, or simulation in simple problems, or something similar will be helpful to elucidate the connection/motivation. Second, the convergence result is weak (ie, condition 4 in theorem 1). Can the authors elaborate on why it’s hard to relax the condition in typical stochastic approximation analysis? Third, some reviewers find the empirical gains to be small, and ask for clarity in confidence intervals in the plots. Finally, the combination of two gradients in line 21 of algo 2 seems ad hoc, and it’s unclear how to choose alpha (related to one reviewer’s comment).

**Audience:**

The topic is interesting to the reinforcement learning community.

**Claims And Evidence:**

The claims are partially supported. The motivation of how removing the max operation mitigates over- and under-estimation in Q-functions is unclear. Experiments on benchmarks like Mujoco show promise, but there are concerns in how to understand confidence intervals (thus how big the gain is). Finally, the theory is limited, given restricted assumptions.

**Resubmission Of Major Revision:**

The authors may consider submitting a major revision at a later time.